# LncRNA NEAT1 in Paraspeckles: A Structural Scaffold for Cellular DNA Damage Response Systems?

**DOI:** 10.3390/ncrna6030026

**Published:** 2020-07-01

**Authors:** Elisa Taiana, Domenica Ronchetti, Katia Todoerti, Lucia Nobili, Pierfrancesco Tassone, Nicola Amodio, Antonino Neri

**Affiliations:** 1Department of Oncology and Hemato-oncology, University of Milan, 20122 Milan, Italy; domenica.ronchetti@unimi.it (D.R.); lucia.nobili@unimi.it (L.N.); 2Hematology, Fondazione Cà Granda IRCCS Policlinico, 20122 Milan, Italy; katiatodoerti@gmail.com; 3Department of Experimental and Clinical Medicine, Magna Graecia University of Catanzaro, 88100 Catanzaro, Italy; tassone@unicz.it (P.T.); amodio@unicz.it (N.A.)

**Keywords:** lncRNA, NEAT1, paraspeckle, DNA damage repair, cancer, neurodegenerative disease

## Abstract

Nuclear paraspeckle assembly transcript 1 (NEAT1) is a long non-coding RNA (lncRNA) reported to be frequently deregulated in various types of cancers and neurodegenerative processes. NEAT1 is an indispensable structural component of paraspeckles (PSs), which are dynamic and membraneless nuclear bodies that affect different cellular functions, including stress response. Furthermore, increasing evidence supports the crucial role of NEAT1 and essential structural proteins of PSs (PSPs) in the regulation of the DNA damage repair (DDR) system. This review aims to provide an overview of the current knowledge on the involvement of NEAT1 and PSPs in DDR, which might strengthen the rationale underlying future NEAT1-based therapeutic options in tumor and neurodegenerative diseases.

## 1. Introduction

Nuclear paraspeckle assembly transcript 1 (NEAT1) is a functionally conserved long non-coding RNA (lncRNA), abundantly expressed in a variety of mammalian cell types. NEAT1 has been described as the essential structural scaffold for the assembling of paraspeckles (PSs), which are membraneless nuclear bodies performing different fundamental cellular functions.

Notably, NEAT1 deregulation has been reported in various types of cancers with important implications in the regulation of apoptotic cell death, cell growth, proliferation, invasion and metastasis, thus suggesting its possible role as a therapeutic target [1,2]. Notably, NEAT1 overexpression leads to an increase in the number and dimension of PSs in tumor cells [3]. Data regarding the impact of NEAT1 and PSs in carcinogenesis and the underlying mechanisms of their deregulation are progressively emerging [2]. In particular, recent experimental data in different types of tumors have suggested the involvement of NEAT1 and/or PSs in the context of DNA damage repair (DDR) systems [3,4,5].

Growing evidence highlights a deregulation of NEAT1 expression also in neurodegenerative diseases, including Huntington’s disease (HD) [6,7], amyotrophic lateral sclerosis (ALS) [8,9] and Parkinson’s disease (PD) [10]. Indeed, although the exact role of NEAT1 remains controversial, impaired NEAT1 expression levels could impact HD affected neurons by altering gene regulation [6]. NEAT1 overexpression, together with an increase in PSs density, has been found in ALS motor neurons, suggesting a direct contribution of NEAT1 in ALS disease by modulating the functions of ALS-associated RNA-binding proteins [8]. Concerning PD, Simchovitz et al. demonstrated neuron-specific formation of NEAT1-based PSs at the substantia nigra of PD patients, as well as the increase of both NEAT1 and PSs in cultured neuronal cells upon oxidative stress induction [10]. Although it is clear that changes in NEAT1 expression and PS assembly are associated with neuronal damage, our understanding of NEAT1 contribution to the disease pathogenesis is still to be elucidated [11]. Interestingly, the findings from different studies raise the possibility that defects in the DDR underlie brain aging and the development of age-related neurodegenerative disorders [12]. Hence, these considerations prompt further investigation on the NEAT1 and PSs roles in DDR also in neurodegenerative diseases.

Herein, we provide an overview of the current knowledge on NEAT1 and PS proteins (PSPs) dynamics, which could be instrumental for a better comprehension of disease pathobiology and for the design of novel NEAT1-based therapeutic options in human diseases.

## 2. NEAT1 as Essential Structural Scaffold for Paraspeckle Assembling

PSs are dynamic and membraneless nuclear bodies identified for the first time in 1993 in HeLa cells as electron-dense structures of the interchromatin granule-associated zone [13]. The definition of PSs became more detailed in 2002, when they were described as nuclear bodies enriched of specific members of the drosophila behavior human splicing (DBHS), a family of RNA binding proteins (also known as PSPs): PSPC1 (paraspeckle protein 1), NONO (non-POU domain-containing octamer-binding protein), SFPQ (splicing factor proline- and glutamine-rich), RBM14 (RNA Binding Motif Protein 14) and CPSF6 (cleavage and polyadenylation specificity factor subunit 6) [14,15,16].

Notably, in 2005, a further level of complexity to this scenario was added when Fox et al. demonstrated that PSs are RNase-sensitive structures that depend on active transcription by RNA Polymerase II (RNA Pol II), suggesting the requirement of RNAs for their maintenance [15]. Some years later, four groups independently discovered that the architectural lncRNA NEAT1 acts as scaffold for PS assembly and demonstrated that depletion of NEAT1 leads to disintegration of these structures [17,18,19,20], concluding that NEAT1 is essential for the assembling and maintenance of these membraneless nuclear bodies [21,22].

Two mono-exonic isoforms (NEAT1_1 and NEAT1_2) are transcribed from the NEAT1 locus, also called familial tumor syndrome multiple endocrine neoplasia (MEN) type I, on human chromosome 11 [23]. The two isoforms share an identical 5′ sequence of 3.7 kb, which constitutes the short polyadenylated NEAT1_1 transcript [21]. The long NEAT1_2 isoform (23 kb) is not polyadenylated and it is produced when hnRNP K (Heterogeneous nuclear ribonucleoprotein K) protein activity masks the 3′-end processing signals for NEAT1_1 transcription. NEAT1_2 is subjected to RNAseP cleavage with the generation of a tRNA-like structure and it forms a triple helix at its 3′ end which has been implicated in the stabilization of the transcript and protection from 3′–5′ exonucleases action [20,24,25] (Figure 1).

Unlike mRNAs, which contain an open reading frame that is defined by the genetic code, lncRNAs have distinct RNA domains that determine their function; indeed, by means of specific RNA motifs, they can fold into complex structures and interact with multiple partners including RNAs, DNA and proteins, thus modulating their activities and functions [26]. Accordingly, it has been reported that NEAT1 binds to numerous genomic sites in human cells, mainly represented by active genes, suggesting that NEAT1 could be considered a regulatory factor for several genes and pathways [27]. Notably, it has also been demonstrated that only the longer isoform NEAT1_2 is responsible for PSs formation and maintenance [28,29,30], whereas the short NEAT1_1 isoform appears to be localized in several non-paraspeckle loci, suggesting a PSs-independent role [31].

### 2.1. Paraspeckles Biogenesis and Essential Factors for Their Assembling

As recently reviewed by Hirose et al., PSs biogenesis can be described as a two-step process: (1) synthesis of individual NEAT1_2-ribonucleoprotein (RNP) complex; and (2) assembly of about 50 NEAT1_2-RNP complexes in a final PS structure, occurring in close coordination with NEAT1_2 transcription [29]. This process requires the presence and binding of seven essential PSPs, belonging to two distinct categories (Category 1A and Category 1B) based on their role [30].

Category 1A includes four proteins that are considered essential in the first step of PS biogenesis: NONO, SFPQ and RBM14, which are able to directly bind to NEAT1_2 variant, stabilizing it and avoiding its possible degradation, and hnRNP K that promotes NEAT1_2 transcription [30,32]. On the other side, FUS (FUS, DNA/RNA-binding protein fused in sarcoma/translocated in liposarcoma), DAZAP1 (DAZ-associated protein 1) and hnRNP H3 (heterogeneous nuclear ribonucleoprotein H3) are essential proteins of Category 1B required in the second step of the process [30,32] (Figure 1).

The analysis of the structure of these seven essential PSPs (Figure 2) revealed that they all have RNA binding motifs; in detail, six of them possess at least one RNA-recognition motif (RRM), whereas the remaining one has three K homology RNA-binding domains (KH) [33]. In addition, FUS has a zinc-finger motif (ZnF), which itself can display RNA binding activity [33,34] (Figure 2). Altogether, these findings suggest that all these PSPs could directly bind NEAT1_2 within PS.

Furthermore, coiled-coil domains in NONO and SFPQ have been reported to form polymers required for PS integrity; indeed, disruptive mutations of the coiled-coil interaction motif result in SFPQ mislocalization, impaired formation of nuclear bodies and total abrogation of SFPQ molecular interactions [34].

Recent data by Fox et al. suggested that PSPs, once in close proximity and bound to their RNA scaffold, oligomerize and recruit additional proteins by means of their low complexity region domains (Figure 2) and mediate a liquid–liquid phase separation process which determines the liquid-like state, characteristic of PSs [35]. Interestingly, the prion-like domain (PLD) subclass of low complexity regions has been found in all essential PSPs except hnRNP K, suggesting a broad network of PLD-mediated interactions within these nuclear bodies [36,37]. In fact, it is well established that PLDs allow proteins to aggregate and drive ribonucleoprotein granule assembling [36].

Finally, the SWItch/Sucrose NonFermentable (SWI/SNF) chromatin-remodeling complex plays a crucial role in organizing the protein–protein interaction network required for intact PS assembly, by linking Category 1A and 1B proteins together with NEAT1_2 [38] (Figure 1). Indeed, BRG1 (SMARCA4), the core component of the SWI/SNF chromatin remodeling complex, and other SWI/SNF subunits (BAF170, BAF155, BAF57 and BAF47) were localized within PSs and their silencing resulted in marked PS disintegration. Furthermore, interactions between SWI/SNF components and essential PSPs were maintained in NEAT1-depleted cells, suggesting that the SWI/SNF complex not only facilitates interactions with PSPs, but also recruits them during PS assembly [38].

### 2.2. Cellular Functions of Paraspeckle

From a functional point of view, PSs were originally considered dispensable organelles due to the observation that NEAT1 knock-out (KO) mice were healthy and fertile [39]. It was later discovered that NEAT1, along with PS, were required for corpus luteum formation and that modulation of NEAT1 expression levels could impair fertility, as well as lactation and mammary gland development in mice [40].

At a molecular level, at least three main cellular functions have been described for PSs (Figure 3). The first one is gene regulation through the adenine (A) to Inosine (I) editing process by the activity of the adenosine deaminase enzyme RNA specific (ADAR) [17]. Edited transcripts are able to bind to a multiprotein complex including NONO, SFPQ and Matrin 3 within PSs, therefore remaining retained within the nucleus [41].

Regarding their second activity, PSs act as molecular sponges for RNA binding proteins and PSPs, which on their own could regulate genes. The sequestration of these proteins within PSs, although not impacting their functionality, affects their ability to interact with their normal target genes, with a consequent loss of function effect [42,43]. Hence, since PSs have been shown to be induced upon specific conditions, like hypoxia and stress, they could modulate gene expression according to different stimuli [3,4,44].

The third PS function is related to microRNA (miRNA) biogenesis. It strikingly depends on NEAT1 ability to promote assembly of the microprocessor complex involved in miRNA processing through the action of SFPQ protein. In fact, NEAT1 actively facilitates interactions between PSs and miRNAs by forming hairpin loops, which are the secondary structures bound by the microprocessor complex [45].

Finally, there is growing evidence (reviewed in the following paragraphs) concerning the biological and molecular effects of NEAT1 and/or essential PSPs silencing in DDR pathway, suggesting an additional role of PSs in the regulation of the DDR system.

## 3. Involvement of NEAT1 in DNA Damage Response

Cells are equipped with excellent DDR systems able to repair various DNA lesion types [46,47]. P53 can modulate virtually all DNA repair processes through both transcription-dependent and -independent mechanisms [48]. In fact, the induction of p53 by a wide range of stress signals, including exposure to DNA damaging agents, increased reactive oxygen species (ROS) or hypoxia, leads to a p53-mediated activation, either directly or indirectly, of the DNA-damage response [49].

NEAT1 has been identified and validated as a p53 target by several studies that also pointed out the impairment of DDR efficiency in NEAT1 depleted cells [3,4,50,51,52]. In detail, in 2015 Blume et al. reported that the induction of NEAT1 in chronic lymphocytic leukemia (CLL) primary cells could play a role in the DNA damage pathway [4], demonstrating that, after DNA damage, NEAT1 is induced in the presence of functional p53 but not in CLL carrying p53 mutation, and that its induction is closely related to cell death upon DNA damage [4]. In human breast cancer cells, Adriaens et al. showed that treatment with the p53-inducer Nutlin-3 induces NEAT1 expression levels in association with an increase in PSs formation; at the same time, they found that NEAT1 silencing leads to accumulation of DNA damage and to induction of DDR signaling, as confirmed by the enhancement in γH2AX (Histone H2AX) levels and DNA damage foci formation, as well as the increase in the phosphorylated fraction of KAP1 (KRAB-associated Protein 1), known to be an ATM (ataxia-telangiectasia mutated kinase) substrate [3]. Furthermore, they reported that ATR (ataxia telangiectasia and Rad3-related protein) signaling was negatively affected in NEAT1 knocked down (KD) cells exposed to hydroxyurea, as confirmed by the decrease of active amounts of checkpoint kinase CHK1 and replication protein RPA32 (replication protein A 32 kDa subunit) [3]. In line with these findings, our group recently found that NEAT1 silencing in human multiple myeloma cell lines (HMCLs) is associated with the disappearance of PSs structures and leads to a significant modulation of genes and proteins involved in DNA repair processes, including homologous recombination (HR) [5]. Focusing on the HR pathway, we found a reduction of both RAD51 (DNA repair protein RAD51 homolog 1) protein expression level and the phosphorylated fraction of kinase proteins CHK1 and CHK2, all of them involved in the DNA damage checkpoint. We also uncovered a down-modulation of the active amount of both RPA32 and BRCA1 (breast cancer type 1 susceptibility protein), which are a DNA-damage sensor and an essential effector of DNA repair mechanisms, respectively. Moreover, NEAT1 down-regulation resulted in increased genotoxic stress [5].

In accordance with these results, we reported, in HMCLs, a synergistic anti-tumor effect by combining NEAT1 silencing and the PARP (poly ADP-ribose polymerase) inhibitor olaparib, known to induce synthetic lethality in tumors with HR deficiency. Furthermore, we demonstrated that the NEAT1 KD effect in HMCLs is synergic with proteasome inhibitors (bortezomib and carfilzomib) or alkylating agent (melphalan) treatments, indicating that NEAT1 silencing triggers a general sensitizing effect to anti-MM (multiple myeloma) drugs. To this regard, we found that bortezomib or carfilzomib reduce the amount of the phosphorylated form of RPA32 (pRPA32) and that the concomitant NEAT1 silencing further decreases pRPA32 levels in HMCLs, highlighting a crucial role of NEAT1 in HR regulation. Importantly, NEAT1 KD showed a significant reduction of cell viability in a HMCL resistant to bortezomib, as compared to the treatment with the proteasome inhibitor, indicating NEAT1 involvement in chemoresistance mechanisms [5]. These results and those by Adriaens et al. [3,5] suggest that NEAT1 and PSs could represent promising targets to improve the efficacy of a wide range of drugs, including p53 reactivating agents. Moreover, the observation that stabilization of p53 induces NEAT1 and PS formation, which prevents the accumulation of excessive DNA damage in cells undergoing replication stress [3], reveals another way through which p53 could preserve genomic integrity. Interestingly, Adriaens et al. recently reported that NEAT1_1 expression levels are different throughout cell cycle phases, with high levels in G0/G1 mainly localized outside PSs, and decreased expression once cells enter the S phase of cell cycle. In contrast, NEAT1_2 expression remains stable throughout the cell cycle, being the only isoform detectable in the S phase [28]. Considering that the HR pathway occurs during the S and G2 phase of the cell cycle, it could be assumed that NEAT1_2 is the main one responsible for the reported molecular effect. In line with this, Adriaens et al. demonstrated that only NEAT1_2 is essential for ATR signaling activation in response to replicative stress [3], also suggesting that NEAT1_1 could represent a nonfunctional transcriptional variant required to keep NEAT1 transcription active, allowing cells to quickly activate a PS-dependent survival pathway when exposed to harmful stimuli [28].

However, despite much evidence of NEAT1 involvement in the DDR systems, the precise molecular mechanism underlying the role of NEAT1 in DDR remains to be clarified.

Based on the huge amount of data concerning the activity of essential PSPs in DDR, it could be hypothesized that NEAT1 involvement in this pathway is indirect and that PSs act as molecular sponges, promoting the retention and sequestration of proteins directly implicated in DDR processes. In this perspective, NEAT1 could represent, within PSs, the possible structural scaffold for the cellular DDR system.

## 4. Involvement of Paraspeckle Proteins in the DNA Damage Response

In the following sections we will summarize the available information concerning the role of each essential PSPs in DDR.

### 4.1. NONO

Together with SFPQ and PSPC1, NONO is a member of the multifunctional DBHS family of proteins that can bind DNA, RNA and proteins [21,53]. Structural and biological data regarding NONO suggest that it rarely functions as single player and its activity is regulated by post-translational modifications and by interaction with other partner proteins [54].

An important role of NONO in double-strand breaks (DSBs) repair has been reported by different authors (Figure 4, Table 1) [55,56,57,58,59,60,61,62]. To this regard, it has been demonstrated that NONO, together with SFPQ, creates an heterodimer that is associated with ATP-dependent DNA helicase 2 KU70/KU80 proteins to form a functional pre-ligation complex with damaged DNA, suggesting that the NONO/SFPQ complex could represent a DNA end-joining factor that cooperates with other proteins known to participate in the non-homologous end joining (NHEJ) pathway (Figure 4) [56]. Further evidence demonstrated that NONO/SFPQ heterodimer directly interacts with Matrin 3, creating a complex with KU70/KU80 and LigIV (DNA ligase 4), all major players in the NHEJ pathway of DSBs repair (Figure 4) [62]. NONO silencing sensitizes HeLa cancer cells to ionizing radiation (IR) and impairs NHEJ signaling both in vitro and in vivo [59]. However, HeLa cells silenced for NONO retain a good viability despite the reduction of NHEJ system efficiency, suggesting a possible feedback mechanism through its homologous DBHS protein PSPC1, whose expression increases upon NONO KD [59]. NONO is a PAR (poly ADP-ribose)-binding protein and its recruitment to DNA damage sites is PAR-dependent; interestingly, using a stable reporter cell line that allows monitoring HR activity, it was found that NONO KD leads to an up-regulation of HR by almost 40%, indicating that NONO positively regulates NHEJ and, at the same time, represses HR [59].

NONO involvement in the HR pathway has been confirmed by Kuhnert at al., demonstrating that, together with SFPQ, it directly interacts with TopBP1 (DNA topoisomerase 2-binding protein 1) and co-localizes at sites of laser-induced DNA damage (Figure 4) [60]. TopBP1 is able to physically associate in vivo with NBS1, which in turn regulates its recruitment to DNA damage sites. TopBP1 silencing has been reported to result in hypersensitivity to IR and Mitomycin C treatment, increased frequency of sister-chromatid exchange level and finally a reduced frequency of DNA DSBs-induced HR repair [63].

In line with these data, Li et al. showed that NONO silencing in HeLa cells leads to a delay in DSB repair and that stable NONO KD cell lines significantly raise IR-induced chromosomal aberrations [61]. Further confirmation has been reported by Alfano et al. showing that NONO silenced cells exposed to UV radiations are still able to synthesize DNA and display an impairment in the phosphorylation of CHK1, with a consequent defective intra-S-phase checkpoint activation in response to UV irradiation [55]. As a molecular mechanism of this effect, the authors reported that NONO promotes the loading of TopBP1, which acts upstream of the ATM and ATR kinases [55], confirming previous data published by Kuhnert et al. [60].

NONO is subjected to RNF8-dependent ubiquitination and proteasome degradation and this mechanism is crucial in a negative feedback loop required to turn off ATR-CHK1 checkpoint signalling in UV-DDR [57]. In conclusion, although NONO is required for ATR-CHK1 signalling [55], it is also negatively controlled in a RNF8- and proteasome-dependent ways [57]. Finally, NONO was also found implicated in chemoresistance, since its silencing sensitizes HeLa cells to treatment with cisplatin [58].

### 4.2. SFPQ

PSF/SFPQ is the second member of the DBHS family of proteins [21,53]. As reported above, DBHS proteins rarely function alone and their activity is regulated by interaction with other protein partners [54]. For this reason it is not surprising that even SFPQ has a direct role in DDR system, specifically in the recognition and repair of DNA DSBs [56,60,62,64,65,66,67,68,69] (Figure 4, Table 1).

As stated above, SFPQ, together with NONO is able to cooperate with KU70/KU80 forming a functional pre-ligation complex with damaged DNA, suggesting its active role in NHEJ (Figure 4) [56,62]. Moreover, SFPQ silencing in U2OS osteosarcoma cells leads to a significant delay in the disappearance of damage-induced nuclear foci formed by the DSB sensor 53BP1 (TP53-binding protein 1), demonstrating that SFPQ is required for timely DSBs repair [62].

As previously reported for NONO, SFPQ/NONO heterodimer directly interacts with TopBP1 and co-localizes at laser-induced DNA damage site, suggesting a possible implication of SFPQ in HR pathway (Figure 4) [60]. An involvement of SFPQ in HR has also been reported by Akhmedow et al., who showed that a peptide sequence isolated from HeLa cells and identical to SFPQ was able to promote DNA strand invasion, possessing the ability to bind not only RNA but also ssDNA (single stranded DNA) and dsDNA (double stranded DNA) and, facilitating the renaturation of complementary ssDNAs [64]. Based on these results, SFPQ could be crucial in HR pathway by promoting the formation of the D-loops structures, which are the products of the homologous pairing reaction between ssDNA fragments and superhelical dsDNA (Figure 4) [64].

SFPQ directly interacts with RAD51, acting both as activator or inhibitor of the RAD51-mediated homologous pairing and strand exchange depending on RAD51 concentration (Figure 4) [66]. In fact, SFPQ promotes both steps when RAD51 is present at low concentrations, but inhibits these reactions at high RAD51 concentrations. Morozumi et al. hypothesized that in the early step of HR, when RAD51 accumulation at the recombination sites is not yet sufficient, SFPQ may function as activator, while, in later steps, when excess of accumulation of RAD51 could lead to an uncontrolled recombination and possible chromosomal aberrations, SFPQ may act as RAD51 inhibitor, removing it from ssDNA and thus suppressing inappropriate recombination reactions [66]. Further evidence showed that SFPQ directly interacts with RAD51D, and that deficiency of both proteins leads to a synthetic lethal effect [67].

Finally, the involvement of SFPQ in DDR has been confirmed by functional approaches showing that SFPQ KD results in complete HeLa cells death upon radiation treatment, whereas the re-expression of SFPQ rescues survival [65]. Notably, the SFPQ region essential for survival rescue is located in the N-terminal domain of the protein, possibly in RRM1 [65], previously identified as responsible for recruitment of SFPQ-containing complexes to sites of dense DNA damage [68,69].

### 4.3. RBM14

RBM14 mediates key interactions with NEAT1 and several other essential proteins within the same subnuclear organelles. RBM14 contains two amino-terminal RRMs domains and a long low-complexity PLD domain (Figure 2) required for its robust interaction with NONO and for PS formation in cell [36]. RBM14 PLD domain is sufficient for PS targeting demonstrating that it is essential for their building and integrity [36].

A first evidence of RBM14 involvement in DDR was reported by Iwasaki et al. showing that RBM14 interacts with DNA-PKcs (DNA-dependent protein kinase catalytic subunit) and KU80 suggesting its direct involvement in the NHEJ pathway (Figure 4, Table 1) [70]. Afterwards, RBM14 KD was found to significantly influence the kinetics of DSBs repair and reduce DNA-PKcs phosphorylation in glioblastoma multiforme (GBM), without affecting the expression level of total DNA-PKcs, and sensitizing radio-resistant GBM cells in vivo [71]. More recently, Simon et al. confirmed the crucial role of RBM14 in regulation of NHEJ upon DNA damage, demonstrating that its presence is essential for efficient recruitment of DNA repair protein XRCC4 and non-homologous end-joining factor 1 XLF proteins to chromatin (Figure 4, Table 1) and for the release of KU proteins from the DNA damage site, therefore avoiding accumulation of DSBs in cells [72]. The same group demonstrated that KU protein directly interacts with RBM14, likely acting as a scaffold protein to link KU-DNA-PKcs and LigIV/XRCC4 complexes. They also ruled out a possible implication of RBM14 in the regulation of HR, suggesting that it specifically controls the NHEJ pathway [72].

### 4.4. HnRNP K

HnRNP K is an evolutionarily conserved factor found in the nucleus and cytoplasm [73]. HnRNP K is considered an important protein of the DDR system as its induction is dependent on the action of ATM and ATR kinases [74]. Furthermore, hnRNP K represents an important p53 co-factor, requested for p53-dependent cell cycle checkpoints and p53-dependent transcriptional induction upon IR-treatment [74]. In addition, hnRNP K was found to be essential for the survival of tumor cells after irradiation; cells with mutant p53 seem to be more sensitive to hnRNP K KD [75]. Overall, *hnRNP K* can be considered a tumor suppressor gene and its involvement in proliferation, genome stability and DDR have been widely confirmed in mouse models [76].

Moreover, considering that p53 is actively involved in all DDR pathways [77], either as a transcription regulator of genes requested for efficient DDR or through direct protein–protein interaction with repair players, it is clear that hnRNP K could have a potential role in both single-strand break (SSB) and DSB repair programs (Table 1).

### 4.5. FUS/TLS

FUS/TLS is a multifunctional hnRNP predominantly localized in the nucleus [78]. The PLD of FUS causes in vitro hydrogel formation, determining the phase separation that is essential for PSs assembling; indeed, such a domain is required for PS formation in cell [79].

FUS/TLS plays a role in multiple aspects of RNA metabolism [80] and in the maintenance of genome integrity [81,82,83,84]. In addition, numerous evidences support its involvement in DDR. For instance, FUS is phosphorylated by ATM and DNA-PK (DNA-dependent protein kinase) in response to DSB-inducing agents [85,86]. Furthermore, FUS is involved in the formation of D-loop structures by promoting the annealing of homologous DNA, which is an essential step in DNA repair by HR (Figure 4, Table 1) [81]. The recruitment of FUS to DNA damage sites precedes the accumulation of γH2AX, and FUS depletion leads to a dampening of DDR, impairing both HR and NHEJ activities [84]. Moreover, FUS directly interacts with HDAC1 (histone deacetylase 1) [84] that has a crucial role in DDR and maintenance of genomic stability [87].

In addition to DNA DSBs repair, FUS is involved also in SSBs repair mechanisms by playing an important role in base excision repair (BER) system. In fact, it is recruited by PARP1 to form a complex that facilitates optimal localization of the DNA repair protein XRCC1/LigIII (DNA ligase 3) at DNA damaged sites (Figure 4) [83].

### 4.6. BRG1 (SMARCA4)

BRG1 encodes an ATPase enzyme that is a catalytic subunit of the SWI/SNF chromatin remodeling complex, frequently mutated in a wide variety of human cancers [88,89]. To be functional, the SWI/SNF complex requires the presence of one of the mutually exclusive ATPase enzymes: BRM or BRG1. The SWI/SNF complex is involved in multiple DNA repair pathways and depletion or inactivation of SWI/SNF subunits, including BRG1, sensitizing cells to DSB-inducing agents by reducing HR and/or NHEJ efficiency [90,91,92,93,94,95,96,97,98,99,100] (Table 1).

It has been demonstrated that the SWI/SNF complex rapidly localizes to DSB, dependently on ATM-mediated signaling and post-translational modifications of histones [95,96]. The precise role of SWI/SNF complexes in DSB repair is still unclear. In fact it has been reported that the SWI/SNF complex promotes efficient damage signaling, as confirmed by the reduction of γH2AX levels early after IR in BRG1 and BRM depleted cells [96]. At the same time, other studies have reported the opposite effect, i.e., increased [94] or persistent [92] γH2AX levels upon BRG1 depletion, lending support to a defective repair. Moreover, BRG1 promotes HR-associated DNA end resection and RPA and RAD51 loading and exchange [97,98]. Accordingly, upon UV exposure, BRG1 depletion reduces ATR-mediated phosphorylation of RPA32, resulting in decreased RPA loading onto chromatin [100], and finally in DDR impairment. Furthermore it has been demonstrated that BRG1 directly interacts with BRCA1 [90,93] and that this interaction is required for BRCA1 recruitment to UV DNA damage sites [100].

### 4.7. DAZAP1

DAZAP1 is a component of the hnRNP family. It co-localizes and physically interacts with the splicing repressor proteins hnRNP A1 (heterogeneous nuclear ribonucleoprotein A1) and hnRNP C1 (heterogeneous nuclear ribonucleoproteins C1) within the nucleus [101].

Evidence regarding the direct involvement of DAZAP1 in DDR is still lacking; conversely, there are experimental data on the role of its interacting partners in DDR. In detail, DAZAP1 and hnRNP A1/A2 are actively involved in the splicing regulation of the defective BRCA1 exon 18, acting as splicing-inhibitory factors [102]. Furthermore, hnRNP A1, an important PSP [30], is actively involved in regulation of the DNA damage-induced splicing response of different mRNAs coding for proteins implicated in apoptotic pathway, cell-cycle regulation and DDR [103].

Concerning hnRNP C1, it coordinates the DDR and radiation-induced apoptosis [104]. Involvement of hnRNP C in DDR has been confirmed by demonstrating that it is a component of a nucleoprotein complex containing PALB2 (partner and localizer of BRCA2), BRCA2 (breast cancer type 2 susceptibility protein) and BRCA1 [105]. Interestingly, interaction between hnRNP C and PALB2 is mediated by RNA [105]. HnRNP C depletion significantly impairs HR, as confirmed by the decrease of BRCA1/2 and RAD51 protein levels [105].

These evidences, together with the knowledge that PSPs often act as a heterodimer, suggest that DAZAP1 could be involved in DDR regulation through still undefined mechanisms (Table 1).

**Table 1 ncrna-06-00026-t001:** DDR system regulated by essential PSPs.

Essential PSP	PSP Category	Role in PS Formation	Molecular Function	DDR Regulated	Reference
NONO	1A	N1_2 stability	RNA BP	HR/NHEJ	[55,56,57,58,59,60,61,62]
SFPQ	1A	N1_2 stability	RNA BP	HR/NHEJ	[56,60,62,64,65,66,67,68,69]
RBM14	1A	N1_2 stability	RNA BP	NHEJ	[70,71,72]
HnRNP K	1A	Inhibition of N1_1 polyadenylation	RNA BP	NER/BER/MMRHR/NHEJ (?)	[74,75,76]
FUS	1B	PS assembly	RNA BP	HR/NHEJ/BER	[81,83,84,85,86]
BRG1		PS assembly	ATPase	HR	[90,92,93,94,95,96,97,98,100]
DAZAP1	1B	PS assembly	RNA BP	HR (?)	[102,103,104,105]
HnRNP H3	1B	PS assembly	RNA BP	N/A	N/A

PSP: paraspeckle protein; PS: paraspeckle; RNA BP: RNA binding protein; HR: Homologous Recombination; NHEJ: Non-Homologous End Joining; NER: Nucleotide Excision Repair; BER: Base Excision Repair; MMR: Mismatch Repair; (?): possible DDR regulated pathway as suggested by literature data.

### 4.8. HnRNP H3

HnRNP H3 is an RNA-binding protein of the hnRNPs family. It is ubiquitously expressed and takes part to create a complex with different nuclear RNAs, playing pivotal roles in a wide range of RNA processing mechanisms, including splicing [106]. To date, no direct involvement of hnRNP H3 in DDR has been reported.

## 5. Conclusions

The crucial role of NEAT1 as structural scaffold for the aggregation and assembly of PSs is well documented [17,18,19,20]. NEAT1 is required to allow interaction and association with essential PSPs possessing RRMs that, in turn, can associate with other proteins through their PLDs, creating flexible and dynamic molecular platforms able to quickly address cellular needs.

In this review, we focused on NEAT1 and PSs involvement in the control and regulation of DDR, which represents a critical step to protect cells against genetic lesions, transcription and replication stress, which could in turn promote genomic instability leading to tumorigenesis and cancer progression [47,107]. Intriguing evidence showed that cellular stress condition, hypoxia, IR and chemotherapeutics, which are all factors known to potentially activate DDR pathways, are also able to induce NEAT1 transcription, PSs formation and their enlargement [3,4,44]. Interestingly, almost all the essential PSPs are actively involved at different levels in the regulation of different DDR pathways, including DSBs repair (HR and NHEJ), as well as the SSBs BER pathway. Hence, PSs enlargement could alter the balance in the cellular distribution and localization of essential PSPs, with the final result of a potential modulation of their functions, including DDR. In this perspective, the structural role of NEAT1 within the PS would suggest an indirect activity of NEAT1 in the cellular DDR system.

However, NEAT1 has been reported as deregulated in neurodegenerative diseases [11] and in the majority of tumors [2] and is known to be a direct target of the transcription factor p53 that, as is common and established knowledge, is defined as the guardian of the genome [3,4,50,51,52]. Based on these considerations, further studies will be important to unravel more precise mechanisms of NEAT1 involvement in genome integrity control and maintenance, which could likely provide a rationale for the development of NEAT1-based therapeutic options in human diseases.

## Figures and Tables

**Figure 1 ncrna-06-00026-f001:**
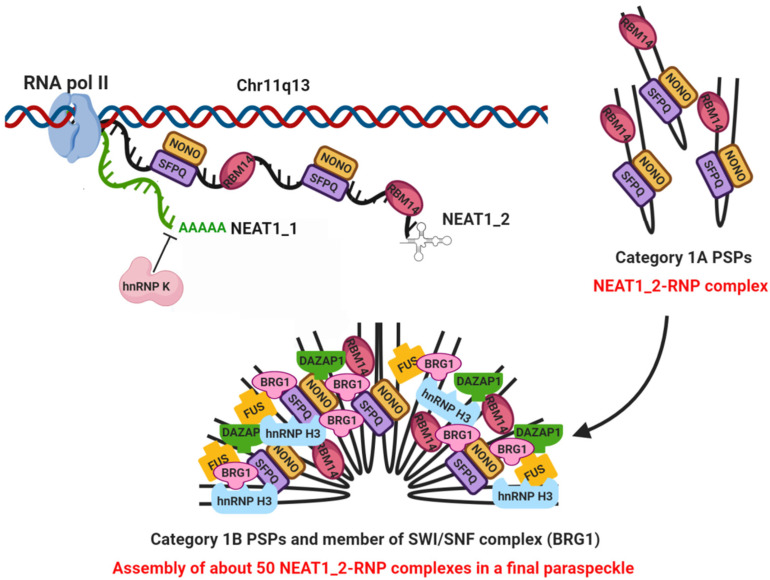
Scheme of the synthesis of NEAT1_1 and NEAT1_2 isoforms and PS biogenesis by the two-step process: (1) synthesis of individual NEAT1_2-Ribonucleoprotein complex with the involvement of Category 1A paraspeckle proteins (PSPs); (2) assembly of about 50 NEAT1_2-Ribonucleoprotein complexes in a final PS structure with the joining of Category 1B PSPs and member of the SWItch/Sucrose NonFermentable (SWI/SNF) chromatin remodeling complex (BRG1).

**Figure 2 ncrna-06-00026-f002:**
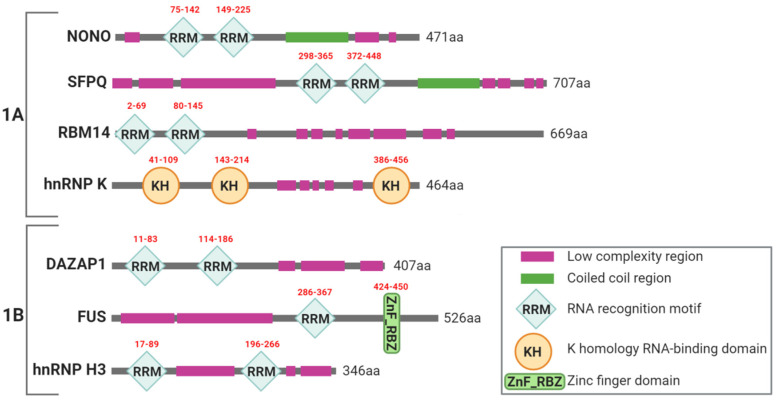
Schematic representation of essential PSPs belonging to category 1A and 1B and their major domains. The amino-acid length of each PSP is shown, whereas the amino-acid interval of each RNA-recognition motif (RRM), K homology RNA-binding (KH) and zinc-finger motif (ZnF) domain is indicated in red upon the corresponding region. Domain legend is shown in the box on the right.

**Figure 3 ncrna-06-00026-f003:**
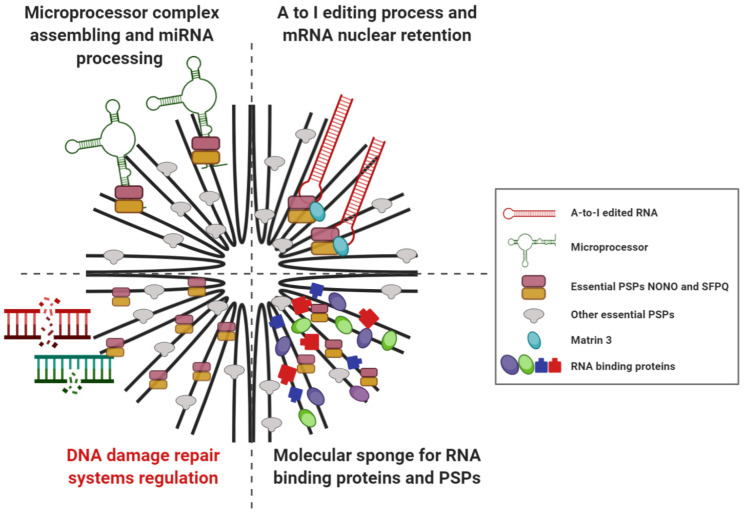
Graphical summary of PSs cellular functions. Upper, right panel: PSs as a gene expression regulator through the A to I editing process and the consequent nuclear retention of different mRNAs. Lower, right: PSs as molecular sponges for RNA binding proteins and PSPs. Upper, left: PSs as regulator of miRNA biogenesis by regulating the assembling of the microprocessor complex involved in processing of miRNA. Lower, left: PSs as possible structural scaffold for cellular DNA damage response systems.

**Figure 4 ncrna-06-00026-f004:**
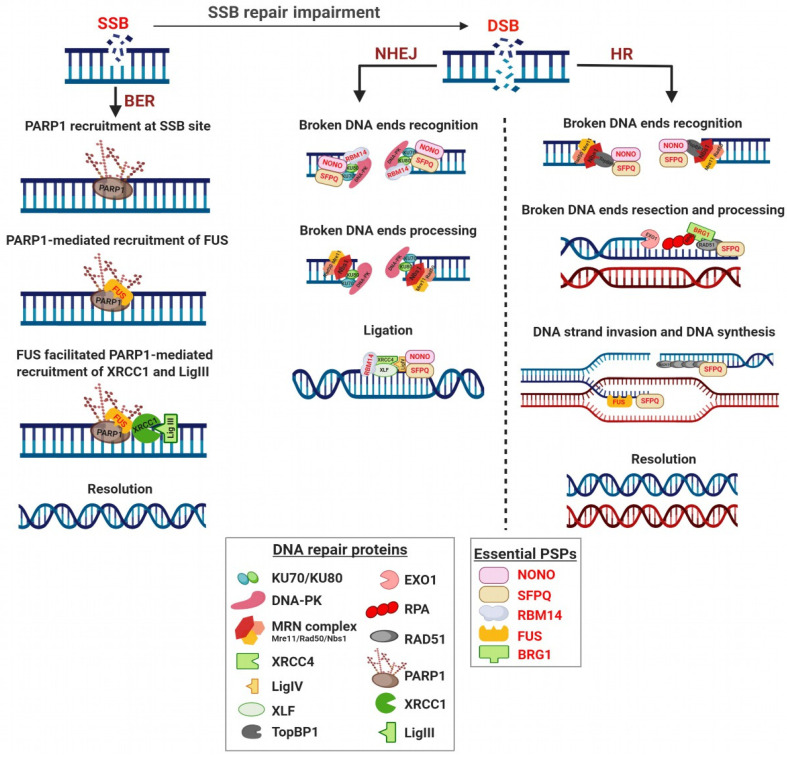
Schematic representation of essential PSPs involvement at different levels of DDR system.

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
