# Peer review of "LncRNA NEAT1 in Paraspeckles: A Structural Scaffold for Cellular DNA Damage Response Systems?"

_ncrna, 2020, doi:10.3390/ncrna6030026_

Round 1
Reviewer 1 Report
The provided review of Taiana et al. describes the role of NEAT1 in paraspeckles and its contribution to DDR. The manuscript summarizes the importance of NEAT1-dependent paraspeckles and the main components concise but comprehensively, which I regard as main aim of a good review. The authors put lots of effort into the structure, style and content of the review and also into the figures. Recent and enough literature has been cited. From my point of view all necessary information is included in this review, and therefore, I do not have a lot of suggestions.
Just some minor comments:
- The introduction is very similar to the abstract, therefore, I suggest to rewrite either the one or the other.
- please check the whole manuscript, if all abbreviations have been fully written at the first mentioning
- as I already said, the figures are well designed and coherent, but in my eyes Figure 3 is overloaded and thus confusing. Maybe the authors could simplify the figure.
- Line 101-103: please explain the importance of PLD in this context, so that the reader can easily follow this paragraph.
Overall, I think this review is well written and provides an important new summary to the scientific community.
Author Response
Referee #1 (Comments to the Author):
The provided review of Taiana et al. describes the role of NEAT1 in paraspeckles and its contribution to DDR. The manuscript summarizes the importance of NEAT1-dependent paraspeckles and the main components concise but comprehensively, which I regard as main aim of a good review. The authors put lots of effort into the structure, style and content of the review and also into the figures. Recent and enough literature has been cited. From my point of view all necessary information is included in this review, and therefore, I do not have a lot of suggestions.
Questions and suggestions:
- The introduction is very similar to the abstract, therefore, I suggest to rewrite either the one or the other.
Prompted by the Reviewer’s suggestion and by comments from the second Reviewer, both Abstract and Introduction has been modified in the revised version of the manuscript.
- please check the whole manuscript, if all abbreviations have been fully written at the first mentioning
Done.
- As I already said, the figures are well designed and coherent, but in my eyes Figure 3 is overloaded and thus confusing. Maybe the authors could simplify the figure.
Prompted by the Reviewer’s suggestion, we modified Figure 3 in the revised version of the review. In detail, in each panel we reduced the number of proteins drawn by specifying only those directly involved in the PSs function under consideration, and merging the others as "other essential PSPs", thus, on the whole, simplifying the image while retaining all information.
- Line 101-103: please explain the importance of PLD in this context, so that the reader can easily follow this paragraph.
Prompted by the Reviewer’s suggestion, we better explained the importance of PLD in the revised version of the paper and added the appropriate reference.
Reviewer 2 Report
The review by Taiana and colleagues summaries available literature on the involvement of an abundant multifunctional lncRNA, NEAT1, and nuclear bodies it assembles, paraspeckles, in DNA damage repair (DDR). The manuscript provides an overview of the paraspeckle structure and function, followed by detailed review of the role for each of the essential paraspeckle components in DDR. Overall, the manuscript is well-written, although careful proof-reading by a native English speaker would be beneficial. My main comment is that the paper suffers from mis-citations (wrong papers cited or relevant papers not cited) as detailed below. This should be corrected, before the manuscript is considered for publication. Secondly, the title suggests that NEAT1 will be mainly discussed within the review, however, two-thirds of the review are devoted to paraspeckle proteins. Therefore, authors may consider revising the title (e.g. removing NEAT1 and keeping paraspeckles only – "Paraspeckles: a structural scaffold…"). Thirdly, whilst I appreciate that the authors have a background in cancer research, it would be beneficial to include a chapter on NEAT1 roles in DDR in neurological/neurodegenerative diseases. Especially, the title suggests that NEAT1/paraspeckle roles in DDR in general, rather than specifically in cancer, are being reviewed. The role of DDR in neurodegeneration is well-established (e.g. see Madabhushi et al., 2014, Neuron), whereas NEAT1/paraspeckle dysfunction is implicated in multiple neurodegenerative diseases (An et al., 2018, Non-Coding RNA Research). There are established links between altered NEAT1 expression specifically in Huntington’s disease (Johnson, 2012, Neurobiology of Disease; Sunwoo et al., 2017, Molecular Neurobiology), ALS (Nishimoto et al., 2013, Molecular Brain; Shelkovnikova et al., 2018, Molecular Neurodegeneration) and Parkinson’s disease (Simchovitz et al., FASEB Journal). Adding this information will allow attracting a wider audience and make the review more well-rounded and comprehensive.
Specific comments:
- Both the abstract and introduction claim that NEAT1 is “highly conserved” however there is little structural conservation even between human and mouse. This should be changed to “functionally conserved” or removed altogether.
- The first part of Introduction (part 1) is vague and is not very well written. For example, the link between NEAT1/PSs and carcinogenesis does not imply that NEAT1 is involved in DDR. It should be made more concise and clear and avoid too generic statements.
- NEAT1 3’ triple helix is described and characterized in Wilusz et al., 2012, Genes Dev and Brown et al., 2012, PNAS (which both should be cited in the review), but not in Ref 10.
- Ref 16 and 17 were not the first to report essential role for NEAT1_2 in paraspeckle assembly, instead, Sasaki et al., 2009, PNAS and Naganuma et al., 2012, EMBO J, should be cited here.
- Line 95 – what is meant by “functional disruptive mutations”?
- The contribution of PLDs in PS assembly was reported in Hennig et al, 2015, JCB and subsequently in Yamazaki et al., 2018, Mol Cell, not in Ref 24.
- Line 124-125 – this has been demonstrated in Ref 7, not 27.
- Line 126 – typo, multiprotein.
- What is Ref 28? Correct ref here is Zhang and Carmichael, 2001, Cell.
- Line 129-131. I cannot agree that sequestration into PS does not affect functionality of RBPs, it is quite the opposite – e.g. SFPQ sequestration into PSs leads to its 50% depletion from nucleoplasm and distraction from promoters of multiple genes – i.e. partial loss of function (albeit without protein degradation and rapid release upon decrease in PS numbers).
- Alongside with Ref 29, Imamura et al., 2013, Mol Cell should be cited.
- Regarding their own findings (Ref 41), could authors discuss the differential role of NEAT1_2 and NEAT1_1 in the p53 pathway and DDR?
- Lines 280 and 311. “In vivo” in this context should be replaced by “in cells” as it is not referring to animal models.
- A new subheading (“4. Involvement of paraspeckle proteins in the DNA damage response”, or similar) should be introduced, because currently proteins (NONO, etc) are discussed under “3. Involvement of NEAT1 in DNA damage response” which is misleading.
- Line 213 – what is “functional” downregulation?
- Line 39. A better reference than Ref 68 can be cited, e.g. an earlier paper by the same group, Vance et al., 2009, Science.
- Line 310 – a Ref is needed for formation of hydrogels by FUS (Ref 59 is for RBM14).
- Ref 70 is about TDP-43 protein, not FUS/TLS.
Author Response
Referee #2 (Comments to the Author):
The review by Taiana and colleagues summaries available literature on the involvement of an abundant multifunctional lncRNA, NEAT1, and nuclear bodies it assembles, paraspeckles, in DNA damage repair (DDR). The manuscript provides an overview of the paraspeckle structure and function, followed by detailed review of the role for each of the essential paraspeckle components in DDR. Overall, the manuscript is well-written, although careful proof-reading by a native English speaker would be beneficial. My main comment is that the paper suffers from mis-citations (wrong papers cited or relevant papers not cited) as detailed below. This should be corrected, before the manuscript is considered for publication. Secondly, the title suggests that NEAT1 will be mainly discussed within the review, however, two-thirds of the review are devoted to paraspeckle proteins. Therefore, authors may consider revising the title (e.g. removing NEAT1 and keeping paraspeckles only – "Paraspeckles: a structural scaffold…"). Thirdly, whilst I appreciate that the authors have a background in cancer research, it would be beneficial to include a chapter on NEAT1 roles in DDR in neurological/neurodegenerative diseases
Especially, the title suggests that NEAT1/paraspeckle roles in DDR in general, rather than specifically in cancer, are being reviewed. The role of DDR in neurodegeneration is well-established (e.g. see Madabhushi et al., 2014, Neuron), whereas NEAT1/paraspeckle dysfunction is implicated in multiple neurodegenerative diseases (An et al., 2018, Non-Coding RNA Research). There are established links between altered NEAT1 expression specifically in Huntington’s disease (Johnson, 2012, Neurobiology of Disease; Sunwoo et al., 2017, Molecular Neurobiology), ALS (Nishimoto et al., 2013, Molecular Brain; Shelkovnikova et al., 2018, Molecular Neurodegeneration) and Parkinson’s disease (Simchovitz et al., FASEB Journal). Adding this information will allow attracting a wider audience and make the review more well-rounded and comprehensive.
We thank the Reviewer for these interesting suggestions.
Concerning mis-citation, we carefully checked all references and amended the reference list as specified point-by-point below.
Regarding the removal of NEAT1 from the title, we agree with the Reviewer that the manuscript has devoted ample space to paraspeckle proteins, however, the review also points out it is NEAT1 itself that plays the central role as paraspeckle scaffold. In addition, NEAT1 seems to be the frontline player to respond to stress stimuli that induce p53 signalling. Overall, the crucial role of NEAT1 emerges alongside the review in both physiological and pathological condition, and it is further underlined in the graphical abstract that introduces the review in the revised version. For these reasons, we believe that the original title better fits with the review message.
Finally, we agree with the Reviewer’s suggestion to include data concerning NEAT1 deregulation in neurological/neurodegenerative diseases and its possible implication in the DDR system, thus making the review more well-rounded and comprehensive. Hence, we added this information in the Introduction of the revised manuscript.
Specific comments:
- Both the abstract and introduction claim that NEAT1 is “highly conserved” however there is little structural conservation even between human and mouse. This should be changed to “functionally conserved” or removed altogether.
Done.
- The first part of Introduction (part 1) is vague and is not very well written. For example, the link between NEAT1/PSs and carcinogenesis does not imply that NEAT1 is involved in DDR. It should be made more concise and clear and avoid too generic statements.
Prompted by the Reviewer’s suggestion and by comments from the first Reviewer, the first part of Introduction has been modified in the revised version of the manuscript.
- NEAT1 3’ triple helix is described and characterized in Wilusz et al., 2012, Genes Dev and Brown et al., 2012, PNAS (which both should be cited in the review), but not in Ref 10.
Prompted by the Reviewer’s suggestion, we included these two references in the revised version of the paper.
- Ref 16 and 17 were not the first to report essential role for NEAT1_2 in paraspeckle assembly, instead, Sasaki et al., 2009, PNAS and Naganuma et al., 2012, EMBO J, should be cited here.
Prompted by the Reviewer’s suggestion, we included Naganuma et al., 2012, EMBO J in the Reference List. We omitted the paper by Sasaki et al., 2009, PNAS, as in the text we were referring specifically to the NEAT1_2 isoform and not to NEAT1 global transcripts.
- Line 95 – what is meant by “functional disruptive mutations”?
We apologize for this editing mistake. In the revised version of the paper we wrote “disruptive mutations of the coiled-coil interaction motif…” meaning that, as a result of these mutations, the domain is no longer able to create interaction, thus losing its functional activity.
- The contribution of PLDs in PS assembly was reported in Hennig et al, 2015, JCB and subsequently in Yamazaki et al., 2018, Mol Cell, not in Ref 24.
Prompted by the Reviewer’s suggestion, the reference has been amended.
- Line 124-125 – this has been demonstrated in Ref 7, not 27.
Prompted by the Reviewer’s suggestion, the reference has been amended.
- Line 126 – typo, multiprotein.
Done
- What is Ref 28? Correct ref here is Zhang and Carmichael, 2001, Cell.
Prompted by the Reviewer’s suggestion, the reference has been amended.
- Line 129-131. I cannot agree that sequestration into PS does not affect functionality of RBPs, it is quite the opposite – e.g. SFPQ sequestration into PSs leads to its 50% depletion from nucleoplasm and distraction from promoters of multiple genes – i.e. partial loss of function (albeit without protein degradation and rapid release upon decrease in PS numbers).
We apologize with the Reviewer for not being fully clear in our statement. Indeed, in the original paper we aimed to underline that PSs recruitment of RBPs affects their ability to interact with their normal target genes; however, despite this sequestration, it should be remembered that RBPs retain the ability to bind and influence promoter regions after being released from PS, meaning that this sequestration is not equal to a loss of function for these proteins. Of course, RBPs sequestration causes a “biological” loss of function effect in the cell. Prompted by the Reviewer’s suggestion, we accordingly modified the text in the revised version of the manuscript.
- Alongside with Ref 29, Imamura et al., 2013, Mol Cell should be cited.
Done
- Regarding their own findings (Ref 41), could authors discuss the differential role of NEAT1_2 and NEAT1_1 in the p53 pathway and DDR?
We thank the Reviewer for this interesting suggestion. Actually, our studies indicated that NEAT1 silencing in human multiple myeloma cell lines affects DNA repair processes, including Homologous Recombination. However, in our experiments NEAT1 knockdown was performed with Gapmers impacting the expression of both NEAT1_1 and NEAT1_2, which prevents us to draw hypothesis.
Concerning this aspect, there are interesting data from Adriaens et al. (Adriaens et al, 2019, RNA), reporting that NEAT1_1 expression levels are different throughout cell cycle phases, with high levels in G0/G1 mainly localized outside PSs, and decreased expression once cells enter the S phase of cell cycle. In contrast, NEAT1_2 expression remains stable throughout the cell cycle, being the only isoform detectable in the S phase. Considering that the HR pathway occurs during the S and G2 phase of the cell cycle, it could be assumed that NEAT1_2 is the main responsible for the reported molecular effect. In line with this, Adriaens et al. demonstrated that only NEAT1_2 is essential for ATR signaling activation in response to replicative stress (Adriaens et al, 2016; Nat Med.), also suggesting that NEAT1_1 could represent a nonfunctional transcriptional variant required to keep NEAT1 transcription active, allowing cells to quickly activate a PS-dependent survival pathway when exposed to harmful stimuli (Adriaens et al, 2019, RNA).
Following the Reviewer’s request, we included this information in the revised version of the manuscript.
- Lines 280 and 311. “In vivo” in this context should be replaced by “in cells” as it is not referring to animal models.
Done
- A new subheading (“4. Involvement of paraspeckle proteins in the DNA damage response”, or similar) should be introduced, because currently proteins (NONO, etc) are discussed under “3. Involvement of NEAT1 in DNA damage response” which is misleading.
Prompted by the Reviewer’s suggestion, data concerning the involvement of paraspeckle proteins in the DNA damage response are reported in a separate chapter in the revised version of the manuscript.
- Line 213 – what is “functional” downregulation?
We agree with the Reviewer that the expression “NONO silencing…”could be more appropriate, and accordingly we modified the text in the revised manuscript.
- Line 309. A better reference than Ref 68 can be cited, e.g. an earlier paper by the same group, Vance et al., 2009, Science.
Done
- Line 310 – a Ref is needed for formation of hydrogels by FUS (Ref 59 is for RBM14).
Prompted by the Reviewer’s suggestion, the reference has been amended in the revised manuscript.
- Ref 70 is about TDP-43 protein, not FUS/TLS.
Ref 70 by Baechtold et al., refers to the nuclear protein POMp75 found to be identical to TLS/FUS, and its ability to promote D-loop formation; hence, we retained this reference as appropriate. We assume that the Reviewer was referring to Ref 71 by Mitra et al., that is about TDP-43 protein and only marginally concerns FUS/TLS. Based on this consideration, we removed Mitra et al., by the Reference List.
Round 2
Reviewer 2 Report
Authors addressed all of my comments.